# Secondary IgA Nephropathy and IgA-Associated Nephropathy: A Systematic Review of Case Reports

**DOI:** 10.3390/jcm12072726

**Published:** 2023-04-06

**Authors:** Maciej Tota, Vanessa Baron, Katie Musial, Bouchra Derrough, Andrzej Konieczny, Magdalena Krajewska, Kultigin Turkmen, Mariusz Kusztal

**Affiliations:** 1Faculty of Medicine, Wroclaw Medical University, 50-367 Wrocław, Poland; 2Faculty of Dentistry, Wroclaw Medical University, 50-435 Wrocław, Poland; 3Department of Nephrology and Transplantation Medicine, Wroclaw Medical University, 50-556 Wrocław, Polandmariusz.kusztal@umw.edu.pl (M.K.); 4Division of Nephrology, Department of Internal Medicine, Meram Medical Faculty, Necmettin Erbakan University, Konya 42090, Turkey

**Keywords:** IgA-associated, gastrointestinal, infection, dermatological, cancer, liver, autoimmune, drug-induced, treatment, pathophysiology

## Abstract

Primary (pIgAN), secondary IgA nephropathy (sIgAN), and IgA-associated nephropathy can be distinguished. While pIgAN has been thoroughly studied, information about the etiology of sIgAN remains scarce. As concerns sIgAN, several studies suggest that different etiologic factors play a role and ultimately lead to a pathophysiologic process similar to that of pIgAN. In this article, we review a vast number of cases in order to determine the novel putative underlying diseases of sIgAN. Moreover, updates on the common pathophysiology of primary disorders and sIgAN are presented. We identified liver, gastrointestinal, oncological, dermatological, autoimmune, and respiratory diseases, as well as infectious, iatrogenic, and environmental factors, as triggers of sIgAN. As novel biological therapies for listed underlying diseases emerge, we suggest implementing drug-induced sIgAN as a new significant category. Clinicians should acknowledge the possibility of sIgAN progression in patients treated with TNF-α inhibitors, IL-12/IL-23-inhibitors, immune checkpoint inhibitors, CTLA-4, oral anticoagulants, thioureylene derivatives, and anti-vascular endothelial growth factor drugs.

## 1. Introduction

### 1.1. Primary IgA Nephropathy

Immunoglobulin A nephropathy (IgAN), also known as Berger’s disease, is the most common autoimmune type of glomerulonephritis [1]. According to a four-hit hypothesis for the pathogenesis of IgAN, the formation of pathogenic IgA1-containing immune complexes (hit 3) is triggered by an increased level of galactose-deficient IgA1 (hit 1) and the production of unique anti-glycan antibodies (hit 2). As a result of the accumulation, the complement system and mesangial cells are activated, leading to the release of cytokines and extracellular matrix proteins (hit 4) [2]. Given the above, chronic inflammation and fibrosis are involved in glomerular function deterioration in IgAN.

The estimated incidence in adults amounts to 25/1,000,000 per year [3]. IgAN is more common among the Asian populations, presented by 45/1,000,000 per year in Japan, than Caucasians (31/1,000,000 per year in France) [4]. Jennette et al. showed that the occurrence of IgAN is approximately 6 times higher in Whites = 7.7% (100/1292) than in African Americans and African Blacks = 1.3% (6/461) undergoing renal biopsy [5]. IgAN can be noticed at any age, wherein perspective and management differ in children and adults [6].

### 1.2. Secondary IgA Nephropathy

Primary IgA nephropathy (pIgAN), secondary IgA nephropathy (sIgAN), and IgA-associated nephropathy can be distinguished. Various etiologies may lead to sIgAN. Amongst them are liver, gastrointestinal, oncological, dermatological, autoimmune, and respiratory diseases, as well as iatrogenic, infectious, and environmental factors. The most common sIgAN triggers are liver diseases, in particular liver cirrhosis [7,8,9].

Whether secondary forms of the disease share common pathways activated by underlying conditions or separate processes that result in comparable pathologic findings remains to be determined. Recent reports suggest that secondary IgAN shares a similar galactose-deficient IgA1-oriented pathogenesis with primary IgAN, at least for IgA vasculitis [10,11]. Wang et al. showed that circulating IgA1 and glomerular IgA1 displayed galactose deficiency of O-glycans in patients with secondary IgAN. Furthermore, no statistically significant differences in plasma Gd-IgA1 or IgA1-IgG complex levels between pIgAN and sIgAN were found [12].

Contrarily, Suzuki et al. and Lee et al. suggested that immunohistochemistry with the KM55 mAb could be an efficient technique for differentiating pIgAN from sIgAN [10,13]. Tang et al. found that pIgAN patients have the highest intensity of KM55 and KM55/IgA ratio and recommend KM55/IgA quantified ratio of 0.78 as the optimal cut-off value to distinguish pIgAN from sIgAN [14]. Nevertheless, recent reports could not confirm its specificity [15]. Furthermore, sIgAN was diagnosed in older patients with a higher Charlson Comorbidity Index (CCI) that less frequently develop hypertension [7,16]. Patients with sIgAN presented higher serum IgA and IgG levels compared to those with pIgAN [12].

Although both sIgAN and IgA-associated nephropathy are derivatives of preceding disease entities, we pinpoint the difference between the two as long as management differs. In sIgAN, renal problems are predominant, significantly deteriorating patients’ life quality, while in IgA-associated nephropathy, alteration in kidneys is not that severe (oligosymptomatic or asymptomatic).

This review evaluates the pathogenic link between primary diseases and sIgAN or IgA-associated nephropathy. Moreover, we summarize case reports published to date in order to demonstrate the outcome of the introduced treatment depending on the IgAN phenotype. The primary purpose of this review was to identify the most common diseases causing secondary IgAN (with the exclusion of IgA vasculitis), focusing on manifestation and outcomes.

## 2. Materials and Methods

We used the Preferred Reporting Items for Systematic Reviews and Meta-Analyses (PRISMA) Statement to conduct this study and registered the protocol on INPLASY (registration code: INPLASY202320022, DOI: 10.37766/inplasy2023.2.0022) [17]. Literature research was performed on 7 October 2022 using three bibliographic databases: MEDLINE, EBSCO, and Embase. We did not set the time frame due to the presence of valuable case reports from several decades ago. Our initial inclusion criteria comprised studies concerning humans written in English. The exclusion criteria were as follows: articles regarding primary IgAN, IgA vasculitis, and data presumed as insufficient or irrelevant.

The keywords “secondary IgA nephropathy” OR “IgA-associated nephropathy” were used to formulate the search string in the first stage of a literature evaluation. We found 856 articles in total: 743 from MEDLINE, 48 from EBSCO, and 65 from Embase. Of those, 290 were rejected: 102 did not meet language criteria, 113 were animal studies, and 75 were duplicates. After screening we singled out 58 applicable case reports.

To ensure the highest accuracy and to find supplemental, relevant case studies, we conducted the second stage of literature research. That comprised a manual evaluation of the references in the articles found at the first stage and searches by independent authors with more specific search strings. That led to the addition of 32 case reports.

After selecting eligible papers and analyzing the available data, we realized that the paper does not strictly meet the definition of a systemic review. Due to limited evidence linking several underlying diseases with sIgAN, we were unable to use the risk of bias assessment tool. Hence, we carried out a critical appraisal of the articles’ quality by criteria similar to Joanna Briggs Institute Critical Appraisal Checklist for Case Reports and decided to still adhere to the Preferred Reporting Items for Systematic Reviews and Meta-Analyses (PRISMA) model to ensure the highest quality of the selected material and scientific content [18]. Two investigators checked the case reports’ quality, and the case reports not meeting the criteria were excluded and marked as “insufficient data”.

Figure 1 depicts the process of literature research. To the best of our knowledge, this is the first conducted systematic review using PRISMA on secondary IgA nephropathy and IgA-associated nephropathy.

## 3. IgA Nephropathy Secondary to Gastrointestinal Diseases

### 3.1. Inflammatory Bowel Disease

Inflammatory bowel disease (IBD) is a disorder characterized by chronic inflammation of the gastrointestinal tract; the two most frequent subtypes are Crohn’s disease (CD) and ulcerative colitis (UC). IgAN is the most prevalent discovery on renal biopsy in IBD and has a much greater diagnostic prevalence than all non-IBD renal biopsies [19]. Hence, that suggested shared pathogenesis [20]. A recent study by Rehnberg et al. demonstrated an increased risk of end-stage kidney disease (EKSD) in IBD-related IgAN [21]. The overall morbidity of CD/UC-related IgAN is prominent.

As concerns UC, Trimarchi et al. suggest that CD4+ T-cell dysregulation and B-cell immunoglobulin oversecretion may be clue factors linking UC and IgAN. In addition, both entities share familiar cytokine profiles, as follows: IL-1, IL-2, IL-5, IL-6, TNF, IFN-γ, and TGF-β [22]. To date, published cases of UC mainly reported IgA-associated nephropathy with no specific renal medication required (Table 1). 

Conversely, a certain number of CD patients needed a surgical procedure performed to alleviate renal symptoms. Regarding the treatment of CD, adalimumab and ustekinumab were reported as putative triggers of sIgAN [25,26]. Thus, we suggest cessation or cautious application of those medications until further studies deny a possible causal relationship.

Clinicians should monitor renal function regularly in patients with inflammatory bowel diseases. Moreover, identifying a connection in common pathophysiology could facilitate the development of a therapy.

### 3.2. Celiac Disease

Several studies have reported an increased frequency of IgAN in patients with celiac disease (CD) [78,79,80]. CD manifests itself by malabsorption, villous degeneration of the small intestine villi, chronic mucosal inflammation, diarrhea, and depression. It develops from an autoimmune response to gliadin, secalin, and hordein found in wheat, barley, and rye. In addition, genetic factors influence susceptibility to CD [81]. Most patients carry the HLA-DQ2/HLA-DQ8 haplotype [82].

Contrary to IBD, the link between IgAN and CD has been proven. Transglutaminase-2 (TG2), IgA, and transferrin receptor 1 (TfR1) are involved in the pathomechanism of IgAN in CD [83]. In CD, TG2 is an enzyme that regulates gliadin deamination and IgA retrotranscytosis across the epithelium layer. Transglutaminase-2 has been found to crosslink IgD, resulting in B-cell receptor (BCR) activation and anti-TG2 autoreactivity [84]. As a result, IgG and IgA are produced as auto-antigens against that protein in patients with CD [83]. Furthermore, TG2 interacts with TfR1, leading to IgA1 deposition and inflammation in the renal mesangium.

Based on Cheung et al.’s study, screening IgAN patients with unexplained gastrointestinal issues for undiagnosed CD would seem appropriate. Until the results are obtained, a gluten-free diet is recommended [85]. An altered diet was also beneficial in sIgAN cases presented [32,33,34,35,36]. Coppo et al. performed a clinical study of patients with IgAN and without evidence of celiac disease. The patients were placed on a gluten-free diet for six months. The authors observed a decrease in circulating IgA immune complexes and reduced proteinuria that were not noticed in the control group restricted from eating meat or dairy. Nevertheless, progression towards renal failure was not slowed down [86]. Therefore, more clinical trials are required to assess whether gluten restriction for all IgAN patients is beneficial.

### 3.3. Dysbiosis

The association between the host and gut commensal microbiota has attracted growing attention recently. Since first described by De Angelis et al. in 2014, a relationship between IgAN and microbiome has been proven by many authors [87,88,89,90]. The gut, salivary, periodontal, subgingival, and tonsillar crypt microbiota are altered in patients with IgAN compared to the healthy group. As a result of the microbiome research, scientists have developed programmed inhibitor cells (PICs) that have antibacterial activity against specific species or strains of bacteria [91]. Thus, potential therapy techniques have emerged.

*Clostridioides difficile* is an opportunistic pathogen that may cause a spectrum of diseases ranging from antibiotic-associated diarrhea to pseudomembranous colitis. These infections often occur due to disturbances in gut microbiota composition, mainly triggered by wide-range antibiotics. Given that, *C. difficile* infections are exponents of dysbiosis. In 1999, Gaughan et al. reported an association between *C. difficile* colitis and sIgAN induced by cefixime. Approximately two months after the first diagnosis, relapse was observed with additional renal function deterioration. A 25-year-old woman presented with subnephrotic proteinuria and gross hematuria. Treatment with methylprednisolone empiric pulse therapy, intravenous cyclophosphamide, and oral vancomycin led to the resolution of symptoms [37].

We believe that receiving thoroughgoing knowledge and conducting clinical trials on the role of the microbiome in IgAN could facilitate the management and therapy of IgAN. Nonetheless, the role of microbiota has been mainly studied as regards pIgAN. Further research is required to determine if a similar association exists in sIgAN patients.

## 4. IgA Nephropathy Secondary to Infections

It is well established that infections can trigger sIgAN. Numerous viral, bacterial, and protozoal diseases have been identified as predisposing factors to the development of IgAN, and many infectious causes have emerged in recent years. The suggested pathomechanism involves specific pathogens or a continuous insult to mucosal infections. Pathogens can contribute to immune complex formation with their GalNAc-containing moieties, which may induce IgG anti-glycan auto-antibody formation.

### 4.1. Viral Infections

Concerning viral infections, HIV, HBV, HCV, and EBV were well described as a trigger of secondary IgAN. Apart from those, several case reports on HAV, HEV, and SARS-CoV-2 have appeared.

#### 4.1.1. Hepatitis B Virus

Hepatitis B virus (HBV) infections are associated with a higher risk of renal injuries. HBs antigen (HBsAg) is pivotal in the pathogenesis of sIgAN [92]. Wang et al. detected a significant difference in clinical and renal biopsy findings in HBsAg-IgAN patients compared to those with primary IgAN. In the HBsAg-IgAN group, a higher rate of IgM and IgG deposition, serum creatinine levels, and 24 h proteinuria were found, corroborating a worse prognosis. The authors discovered that patients treated with RAAS inhibitors had a more significant percentage of renal function survival. Moreover, they recommend incorporating antiviral therapy in cases requiring immunosuppressive treatment [93].

#### 4.1.2. Hepatitis A Virus

Hepatitis A virus (HAV) infections are rarely described as predisposing to sIgAN due to the self-limiting course in the majority of patients. In the case reported by al-Homrany, acute renal failure, nephrotic syndrome, and biopsy-proven IgAN following an HAV infection were observed [94]. Han et al. described a spontaneous remission of IgAN associated with HAV resolution. They suggest that impairment of immune complex clearance related to Kupffer cell damage leading to the formation of mesangial IgA deposits may be a common mechanism responsible for glomerulopathy in hepatic injuries [95].

#### 4.1.3. Human Immunodeficiency Virus

Many authors have proved an association between human immunodeficiency virus (HIV) infection and IgAN. The post-mortem study by Beaufils et al. showed diffuse IgA mesangial deposits in 7.8% (9/116) of AIDS patients [96]. An increased IgA binding to non-mesangial collagen types I to III mediated by fibronectin is observed in HIV-infected patients [97]. In addition, a higher plasma IgA is found in those cases. That results in increased deposition of IgA1 complexes and IgAN development. Regarding treating IgAN secondary to HIV infection, Miyasato et al. described that a combination of methylprednisolone pulse therapy with oral prednisolone and tonsillectomy is successful [42].

Furthermore, Kanno et al. reported that patients infected with non-HIV retroviruses are more likely to develop IgAN, suggesting a similar pathomechanism [98].

#### 4.1.4. Epstein–Barr Virus

Jennette et al. showed that the occurrence of IgAN is approximately 6 times higher in Whites = 7.7% (100/1292) than in African Americans and African Blacks = 1.3% (6/461) undergoing renal biopsy [5]. The explanation for such a disparity has remained unknown for years. Zachova et al. believe that Epstein–Barr virus (EBV) infection is involved in IgAN pathogenesis and racial distribution. EBV was found predominantly in surface IgM- and IgD-positive cells among healthy African Americans. Crucially, most African Blacks and African Americans acquire EBV infection within two years after birth. The IgA system is physiologically insufficient in that period of life, manifested as low serum IgA levels and few IgA-producing cells. Accordingly, EBV infects cells secreting immunoglobulins other than IgA. In comparison, Whites are infected by EBV principally in adolescence. As a result, excessive production of abnormal IgA1 is more pronounced in the White population, leading to a higher occurrence of IgAN. Moreover, the authors implicate that an abnormal expression of homing receptors in EBV-infected B cells with a predilection towards the tonsils and upper respiratory tract may contribute to the pathogenesis of IgAN [99].

#### 4.1.5. Severe Acute Respiratory Syndrome Coronavirus 2

As regards SARS-CoV-2, a few cases of associated IgAN were described. In the case report described by Huang et al., the virus acted as a trigger for exacerbating preexisting IgAN [100]. Farooq et al. evaluated concluded above association. Authors suggest that the elevation of IL-6 levels as a result of mucosal infections, e.g., SARS-CoV-2, leads to abnormal glycosylation of IgA1 antibodies, creating immunological complexes with IgG autoantibodies and depositing them in the tissues. The other cytokines generated in COVID-19, such as IL-1 and TNF, can also potentially lead to the proliferation and maturation of IgA1-producing B cells, thereby leading to IgAN. Further studies corroborating the role of SARS-CoV-2 infection in sIgAN pathogenesis are required [101].

### 4.2. Bacterial Infections

#### 4.2.1. Methicillin-Resistant Staphylococcus Aureus (MRSA)

A case of MRSA-related IgA and C3 codominant crescentic glomerulonephritis is presented in the following: A 64-year-old man developed gross hematuria after a fall. A laboratory workup revealed elevated creatinine, and his blood cultures were positive for MRSA. A renal biopsy revealed IgA-C3 dominant crescentic glomerulonephritis with early changes of diabetic nephropathy. Following treatment for his bacteremia, his creatinine level decreased. During a follow-up evaluation, his urine was negative for proteinuria and hematuria. The pathophysiology behind this phenomenon could involve staphylococcal superantigens that selectively trigger T cells which release cytokines that induce class switching, resulting in selective polyclonal induction of IgG and IgA [68].

#### 4.2.2. Methicillin-Sensitive Staphylococcus Aureus (MSSA)

An 87-year-old patient with a recent history of MSSA infection was found to have a 24 h urine protein of 5 g. He was administered i.v. fluids and a biopsy was performed. He was diagnosed with IgAN and started on steroids, which led to improved laboratory values [69].

#### 4.2.3. Mycoplasma Pneumoniae

In the case of an 18-year-old man, the authors discussed the possibility of a connection between *Mycoplasma pneumoniae* infection and IgAN. The patient did not display any proteinuria or hematuria on previous routine examinations. However, following pneumonia caused by *Mycoplasma pneumoniae*, he developed proteinuria and gross hematuria. The above prompted a kidney biopsy, which revealed IgAN. He was started on erythromycin, which improved his condition. After three months, his renal function entirely recovered from the infection. It could be speculated that the patient’s IgAN resulted from increased serum anti-*M. pneumoniae* IgA antibody concentration. This assumption is supported by the evidence that his serum antibody titer to *M. pneumoniae* drastically was decreased by anti-IgA antiserum therapy [70].

Another case concerning *Mycoplasma pneumoniae* involves a 12-year-old boy whose previous routine urinalysis was normal. He presented with gross hematuria and oliguria after developing severe symptoms of an *M. pneumoniae* infection. Based on a kidney biopsy, it was determined that he developed IgAN, which the authors presumed to be secondary to the infection. The above is supported by the fact that four months after the initial admission, the patient was free of symptoms besides microscopic hematuria [71].

#### 4.2.4. Borrelia Burgdorferi

A 40-year-old man with a history of microscopic hematuria was infected with *Borrelia burgdorferi* after a tick bite. Deterioration of kidney function, evident by the sudden appearance of gross hematuria and proteinuria, prompted the acquisition of a kidney biopsy which showed findings in concordance with IgAN. He was diagnosed with acute disseminated Lyme disease and treated with doxycycline, improving his kidney function. The rapid deterioration of the patient’s renal condition shortly after his infection and its improvement upon treatment of the infection suggest that Lyme disease may cause IgAN or exacerbate previously present IgAN [72].

#### 4.2.5. Bartonella Henselae

In a pediatric case of a 13-year-old boy, IgAN seems to have developed in response to a severe infection with *Bartonella henselae*, a spirochete that causes cat scratch disease. A routine urinalysis two months prior to his infection was negative. However, throughout the course of his infection, he developed hematuria and proteinuria. He received antibiotics and eventually recovered, besides still having microscopic hematuria. Six months after his initial presentation, laboratory investigations detected hematuria and proteinuria. A kidney biopsy was performed to unveil the origin of the abnormal urinalysis, and he was diagnosed with IgAN. It was suggested that the patient might have a genetic predisposition, which in combination with the infection, resulted in an amplified IgA stimulus and its deposition in the kidneys, resulting in IgAN [73].

#### 4.2.6. Osteomyelitis

A 70-year-old patient with chronic osteomyelitis presented with microscopic hematuria and proteinuria. He had no prior history of kidney disease, and his urinalysis prior to developing osteomyelitis was unremarkable. Biopsy was diagnostic for IgAN. Following an amputation, the urinalysis improved with the resolution of proteinuria [74].

#### 4.2.7. Tonsillitis

A 34-year-old patient had a positive strep test and presented with hematuria. Her medical history was remarkable for previous episodes of recurrent hematuria preceded by tonsillitis or pharyngitis. A kidney biopsy was consistent with IgAN and a tonsillectomy was performed. Since then, no further episodes of hematuria have been noted [75]. Tonsillectomy is an effective treatment option for patients suffering from IgAN secondary to recurrent acute tonsillitis. This is due to the fact that during infection, the tonsils increase the production of underglycosylated IgA1 molecules which eventually deposit in the kidneys.

#### 4.2.8. IgA-Dominant Postinfectious Glomerulonephritis

In a review by Nasr and D’Agati, 49 case reports of patients with IgA-dominant postinfectious glomerulonephritis were analyzed. In the vast majority of patients, the etiology factor was found to be the genus *Staphylococcus* [102]. Moreover, *Ureaplasma urealyticum, Escherichia coli, Mycobacterium tuberculosis, Moraxella catarrhalis, Mycoplasma hominis, Haemophilus parainfluenzae*, and genus *Streptococcus* were also detected in a study by Huang et al. [103]. The patients presented with severe renal failure, hypertension, proteinuria, of which roughly half of the cases were in the nephrotic range, and hematuria. Most of the patients were treated with antibiotics, and several were also treated with corticosteroids. However, the overall prognosis in those patients was unfavorable. This is most likely due to the patients having comorbidities, namely diabetes. In the majority of cases, while the disease progressed to ESRD, underlying diabetic glomerulosclerosis was present, which might have also led to such disease progression [102].

While the pathomechanism of IgAN secondary to bacterial infections is not fully understood up to the present time, some authors have suggested that staphylococcal superantigens may play a role [68]. In addition, risk factors such as advanced age and comorbidities such as diabetes also seem to contribute to the development of the disease.

### 4.3. Protozoal Infections

#### Plasmodium Falciparum

A 49-year-old man with diabetes mellitus was infected with *Plasmodium falciparum* and developed biopsy-proven IgAN. The previous urinalysis was unremarkable. However, upon admission due to fever, his urine dipstick displayed microscopic hematuria and proteinuria, and a peripheral blood smear confirmed a *Plasmodium falciparum* infection. Once he was cleared of the infection, his laboratory values normalized, suggesting an association between *Plasmodium falciparum* and IgAN. The authors suggested that *Plasmodium falciparum* may have increased the production of anti-Gal, as anti-Gal antibody concentration was found to be increased in individuals from malaria-endemic countries and patients with *P. falciparum* infection. Those antibodies could result in the formation of aberrantly glycosylated polymeric IgA1, which plays a role in IgAN [76].

### 4.4. Schostosoma Mansoni

A 36-year-old woman with a history of hepatosplenic schistosomiasis was found to have proteinuria on routine evaluation. Based on a biopsy, she was diagnosed with IgAN that developed secondary to the schistosomal infection. *Schistosoma* spp. is a pathogen whose intermediate hosts are Biomphalaria freshwater snails. *Schistosoma mansoni* can cause schistosomal glomerulopathy in the course of chronic exposure [77].

Due to the limited reports of sIgAN following protozoal infections, no definite pathomechanism has been established yet, and further studies have to be conducted.

## 5. IgA Nephropathy Secondary to Autoimmune Diseases

### 5.1. Sjögren’s Syndrome

Sjögren’s syndrome (SS) is a systemic autoimmune disorder characterized by an inflammation of the lacrimal and salivary glands resulting in dryness involving the eyes and mouth. SS can be presented as primary or associated with other connective tissue disorders [60]. Fewer than 10% of cases have been reported in regard to renal involvement in primary SS. The glomerular involvement secondary to primary Sjögren’s syndrome has been outlined in the past only in individual case reports in which membranous nephropathy and membranoproliferative glomerulonephritis have been presented.

Tsai et al. report a case of a 65-year-old female presenting with primary SS with cutaneous leukocytoclastic vasculitis and sIgAN. In this case, the patient presented with a nephrotic range, and despite hemodialysis, the patient’s renal function did not improve [60].

### 5.2. Spondyloarthritis

Spondyloarthritis (SpA) represents the group of inflammatory diseases defined by the inflammation within axial joints and peripheral arthritis, enthesitis, and dactylitis. The development of SpA is determined by genetic factors and explicitly linked with the presence of HLA-B27 [104]. The link between IgAN and spondyloarthropathies has been well established, predominantly in cases of ankylosing spondylitis (AS). Although both diseases have been associated, the pathophysiological pathways interlacing the two remain unclear [61].

Jacquet et al. have reported a case of IgAN associated with ankylosing spondylitis. They highlighted the fact that the use of the anti-TNFα agent infliximab is a potent therapy for treating the rheumatological symptoms of AS; however, it does not certainly control the sIgAN, which leads to the conclusion that the mechanisms of AS and the AS-associated IgAN are unrelated.

Zhang et al. have suggested that the renal function decline among AS patients was substantially associated with an absence of HLA-B27, thus suggesting the importance of being more attentive toward HLA-B27-negative AS patients presenting with sIgAN [105].

### 5.3. Behcęt’s Disease

Behcęt’s disease (BD) is an inflammatory condition in which renal manifestations are not common but do at times occur. Secondary IgAN may occur due to BD, although it is challenging to establish a causal relationship between the two as the cases may also be fortuitous.

Yver et al. described a 46-year-old female who experienced proteinuria and was thus referred for further examination. A diagnosis of BD was established fifteen years prior to the consultation when she had suffered from ulcers in the genital and oral areas as well as arthralgia. The patient exhibited erythema nodosum and uveitis at the consultation, and histopathological analysis obtained from the kidney biopsy detected IgA deposits. Since the renal manifestations have occurred after the diagnosis of BD, particularly following a flare of the original disease, the authors speculated that the IgAN is likely secondary to the BD [106].

Hemmen et al. published a case with a 25-year-old Turkish male who developed IgAN as well as BD. The authors emphasized that it is uncertain whether there was a causal relationship between IgAN and BD or just a coincidence [107].

A Brazilian man with BD was also found to have IgAN. The patient experienced renal manifestations including proteinuria, hematuria, and dark urine, and an eventual renal biopsy detected IgA deposits. However, the authors concluded that it is not possible to eliminate the option that this was a case of primary IgAN in a man with BD [108].

### 5.4. Henoch–Schönlein Purpura

Henoch–Schönlein purpura (HSP) is the most prevalent type of vasculitis which presents in childhood in childhood. However, it may also be present among adults. Clinical manifestations of HSP are believed to be deriving from the IgA deposits in blood vessel walls in the affected organs, including skin, gastrointestinal tract, joints, and kidneys. As regards the treatment of renal involvement, mycophenolate mofetil or cyclosporine is considered the most preferable management [109].

Lee et al. reported a 29-year-old female with HSP who presented with end-stage renal disease and obtained renal transplantation eventually followed by loss of the allograft after five years due to recurrent HSP nephritis. In this case, the authors have concluded that four cycles of plasmapheresis (PA) successfully managed proteinuria and reduced IgA deposits in the course of recurrent HSP nephritis [67].

The coexistence of IgAN with autoimmune diseases has been observed. However, the pathomechanism connecting the two conditions remains unclear, thus requiring further studies. Moreover, understanding the pathophysiology would help to establish the appropriate management of the IgAN related to autoimmunological diseases

## 6. IgA Nephropathy Secondary to Oncological Diseases

The link between cancer and glomerular diseases remains scarce. The occurrence of paraneoplastic glomerulonephritis is rare, and the prevalence is somewhat higher among the elderly. Lung and gastrointestinal tract carcinomas are considered to be the most common types linked with paraneoplastic glomerular disease. Other malignancies that have been correlated with sIgAN include renal cell carcinoma, cutaneous T-cell lymphoma, Hodgkin’s disease, and non-Hodgkin lymphoma. Membranous glomerulonephritis has been associated with solid tumors. Nonetheless, paraneoplastic IgAN has been reported seldomly [110].

The molecular pathomechanisms of carcinomas associated with nephropathy are not well understood. One of the suggested mechanisms that possibly explain the occurrence of nephropathy in patients with malignant disease is the deposition of tumor-specific antibodies or antibody complexes in the glomeruli. The accumulation of immune complexes may lead to a cascade of events, including inflammation, the release of reactive oxygen species, and complement activation, leading to possible glomerular damage [111].

### 6.1. Renal Cell Carcinoma

Renal cell carcinoma is the sixth leading cause of death related to cancers. The male population is at a greater risk of RCC than females, defined by the 2.3 ratio with a median age of 65 years for both males and females [112]. Immunohistochemical analysis of resected kidneys revealed that 11 out of 60 (18%) patients suffering from renal cell carcinoma have additionally demonstrated IgAN. Nevertheless, the mechanism and the possible connection between IgAN and RCC remain obscure.

Mimura et al. have reported three cases of IgAN linked with renal cell carcinoma in elderly patients. They revealed that mesangial IgA deposition is likely caused by the infiltrating plasma cells around renal cell carcinoma production of IgA [62].

### 6.2. Carcinomas of the Gastrointestinal Tract

Carcinomas of the gastrointestinal tract are thought to be among the most common types of malignancies linked with paraneoplastic glomerular disease alongside respiratory tract carcinomas. The gastrointestinal and respiratory tracts produce both isotypes subclass of IgA (IgA1 and IgA2). When malignant cells invade the intestinal mucosa, this leads to an increase in the circulating IgA levels, consequently leading to the formation of mesangial IgA deposits [113].

Kocyigit et al. described the first case of paraneoplastic IgAN associated with recurrence of gastric adenocarcinoma in a 58-year-old male. Interestingly, they have concluded that IgAN can precede the relapse of primary disease; thus, in the cases of malignancies, IgAN should be taken into account [63]. Increased IgA concentrations in the serum and whole saliva of patients suffering from neoplastic disease have been reported. Nevertheless, the correlation between the secretory immunoglobulins of those patients and the progress of their disease remains unknown.

According to the study conducted by Brown et al. investigating the association of the IgA levels of serum and whole saliva with the progression of oral carcinoma, it has been highlighted that primary oral and laryngeal cancer patients had a double increase in serum and salivary IgA levels compared to controls [114].

SIgAN has also been connected with squamous cell carcinoma of the tongue, as presented in Table 1 [115].

### 6.3. Myeloproliferative Neoplasms

Glomerulopathies have infrequently been described in patients suffering from myeloproliferative neoplasms (MPNs) [116]. MPNs are clonal disorders defined by an increase in the production of mature blood cells, which can contribute to an undefined glomerulopathy [117,118]. 

Said et al. have performed a study in order to characterize the glomerular diseases associated with myeloproliferative neoplasms. An evaluation was performed on features of 11 patients’ myeloproliferative neoplasm-related glomerulopathy: of those, eight patients had primary myelofibrosis, and each one with chronic myelogenous leukemia, polycythemia vera, and essential thrombocythemia. They have concluded that glomerulopathy shows to be a late complication of myeloproliferative neoplasms, especially primary myelofibrosis with a guarded prognosis [118].

### 6.4. Hodgkin’s Lymphoma

SIgAN in patients with Hodgkin’s lymphoma is scarce, and the pathomechanisms remain unclear. The link between mixed-cellularity type Hodgkin’s lymphoma and glomerular disease is well acknowledged. The most prevalent histological finding is minimal change nephropathy; others include membranous nephropathy, focal segmental glomerulosclerosis, mesangiocapillary glomerulonephritis, anti-glomerular basement membrane nephritis, and acute necrotizing glomerulonephritis [119].

IgAN could either be present simultaneously with Hodgkin’s disease or precede the onset of Hodgkin’s disease. Previous studies have shown that patients with secondary IgAN associated with Hodgkin’s disease had mixed-cellular type nodular sclerosing [120].

According to the case study published by Yoo et al., the simultaneous occurrence of both Hodgkin’s disease and IgAN suggest a possible pathogenic link between the two conditions. In this case study, the authors reported a case of IgAN in a 60-year-old male with a recent diagnosis of Hodgkin’s lymphoma [64].

While the knowledge about the exact pathomechanism interlacing IgAN and oncological diseases remains scarce, it is worth acknowledging that patients with oncological diseases might have a predisposition to developing IgAN as a complication; hence, further studies are required to properly understand the mechanism connecting the two diseases.

## 7. IgA Nephropathy Secondary to Dermatological Conditions

The literature on dermatological conditions as secondary causes of IgAN is limited. The few case studies published to date are summarized in this review.

### 7.1. Psoriasis

The majority of the case reports between secondary IgAN and dermatological conditions include psoriasis.

Ochi et al. described a case in which a 60-year-old male developed IgAN secondary to psoriasis. Adalimumab was effective for three years until the patient showed signs of deterioration and was admitted to the hospital. The patient’s renal manifestations included nephritis, hematuria, and proteinuria. Treatment constituted discontinuation of the adalimumab as well as a combination of antibiotics, etretinate, an ointment consisting of a steroid, and UV therapy. Upon improvement of the patient’s condition, secukinumab, an IL-17A-specific monoclonal antibody, was implemented. The patient found relief from dermatological symptoms, and the nephritis also resolved. Unfortunately, about five months later, presumably due to a change in the dosing interval of secukinumab, the patient’s condition deteriorated once again, and he experienced nephrotic syndrome. After a kidney biopsy, a diagnosis of IgAN secondary to psoriasis was made. The patient underwent tonsillectomy, steroid pulse therapy, and continuation of the secukinumab leading to the patient’s state improvement [43].

Sakellariou et al. reported two patients with IgAN secondary to psoriatic arthritis. In the first patient, a 52-year-old man, infliximab led to massive clinical improvement in psoriatic arthritis and renal manifestations. The second patient, a 46-year-old man, also initially responded well to the infliximab addition. Unfortunately, in the 46-year-old, there was a decline in the effectiveness of infliximab in the subsequent year. The incorporation of both methotrexate and cyclosporin into the infliximab was successful [44].

A recent case from 2020 described a 28-year-old woman with generalized pustular psoriasis and subsequent IgAN. The clinical course of the IgAN was aggravated by anti-TNF treatment. An improvement of the IgAN was witnessed after the administration of corticosteroids, an IL-17A inhibitor, and tonsillectomy [46].

In a case report by Zadrazil et al., a patient with psoriasis vulgaris also was reported to have IgAN. The patient’s renal and dermatological condition improved after prednisone treatment was implemented [47].

In a study by Dervisoglu et al., one out of eight patients with psoriasis who had renal biopsies done was found to have IgAN. The patient was a 50 year-old-female with subnephrotic level proteinuria [121].

### 7.2. Dystrophic Epidermolysis Bullosa

IgAN has also been witnessed in dystrophic epidermolysis bullosa, in which there are dominant and recessive subtypes.

There have been cases of IgAN reported in recessive dystrophic epidermolysis bullosa. Farhi et al. documented four cases with a biopsy of the renal specimen helping to diagnose IgAN [122]. Ahmadi and Antaya described a case of a 29-year-old female with recessive dystrophic epidermolysis with a severe course of the disease progressing to the development of IgAN. The patient progressed to end-stage renal disease and ultimately required peritoneal dialysis [123].

Kawaski et al. were the first to report dominant dystrophic epidermolysis and subsequent IgAN. The patient had dermatological manifestations of dominant dystrophic epidermolysis from birth and was diagnosed when she was seven months old. The patient suffered from infections throughout her life, and renal symptoms were noted when she was 17. Pathological specimens sent for biopsy confirmed IgAN. After treatment, which included steroids, the renal ailments were resolved [49].

Additionally, a link has been established between non-Herlitz junctional epidermolysis bullosa and severe recessive dystrophic epidermolysis bullosa in the published case reports, as shown in Table 1 [48,50].

Tammaro et al. described a patient with a Hallopeau–Siemens form of epidermolysis bullosa. The patient had experienced hematuria and proteinuria, and a renal biopsy revealed IgA deposits in the mesangium. Due to a decline in renal function, hemodialysis was implemented, stabilizing the renal state [124].

### 7.3. Juvenile Dermatomyositis

Mantoo et al. reported a case of a 10-year-old boy who was diagnosed with juvenile dermatomyositis and IgAN. The patient experienced subnephrotic level proteinuria and microscopic hematuria. Hence, a renal biopsy was performed with the discovery of IgA deposits. The renal manifestations resolved after the implementation of i.v. methylprednisolone and oral prednisolone [125].

As mentioned above, sIgAN can be associated with dermatological conditions such as psoriasis, dystrophic epidermolysis bullosa, and juvenile dermatomyositis. The reported cases shed light on the potential of nephrological involvement arising from conditions that mainly affect the skin. Further studies are required to enhance the current knowledge.

## 8. IgA Nephropathy Secondary to Liver Diseases

It is well established that secondary IgAN can occur following liver diseases. Of these liver diseases, the ones with an alcoholic nature are particularly prevalent.

### 8.1. Alcoholic Liver Conditions

Alcoholic liver conditions can lead to sIgAN [126]. Two patients with alcoholic cirrhosis were found to have histopathological changes consistent with secondary IgAN. The renal histopathological abnormalities in both patients disappeared after steroids were administered for one year [51]. Another two men with alcoholic liver cirrhosis aged 55 and 37 also developed sIgAN. Steroid therapy in both men also led to the resolution of their renal symptoms [52].

The link has been further established by case reports in which cirrhosis led to sIgAN [127,128].

### 8.2. Wilson’s Disease

Shimamura et al. reported a case of sIgAN in a man diagnosed with Wilson’s disease. The 20-year-old found relief from his renal symptoms after using trientine HCl and zinc acetate [53].

IgAN secondary to Wilson’s disease has also been reported in the pediatric population. A 9-year-old developed secondary IgAN with gross hematuria and proteinuria. The introduction of D-penicillamine leads to the remission of hematuria and proteinuria [54].

### 8.3. Hemochromatosis

Nakayama et al. described a patient with primary hemochromatosis who developed IgAN. The 55-year-old man had a severe disease course with a rapid decline in his renal function [55]. Steroid treatment improved the renal state, lowering the serum creatinine.

Prior to this patient, Gouet et al. also described a patient with hemochromatosis and subsequent sIgAN [129].

### 8.4. Viral Hepatitis

Viral etiologies, such as hepatitis B and C of chronic liver disease, can be the culprit of sIgAN [95].

Denha et al. published a case in which a 55-year-old male was diagnosed with sIgAN. The patient had a complicated history which included alcoholic cirrhosis as well as hepatitis C. The authors concluded that the sIgAN developed due to HCV reactivation [56].

Two further case reports connecting hepatitis C and sIgAN concluded that the use of antivirals such as interferon alpha and ribavirin led to the improvement of the patient’s state [57,58].

Han et al. reported a case of IgAN secondary to cirrhosis from hepatitis B virus. This patient also experienced acute renal failure and gross hematuria. Lamivudine implementation led to the complete remission of renal symptoms [130].

Francisco et al. described an intriguing case in which a patient who received a renal allograft developed IgAN two years afterward. One year prior to the nephrotic-range proteinuria, the patient had experienced chronic active hepatitis due to hepatitis C. It is possible that the IgAN was secondary to hepatitis as a renal biopsy taken two weeks post-transplant revealed no changes, which would be consistent with IgAN [131].

### 8.5. Autoimmune Hepatitis

Chen et al. reported two patients with Sjogren’s syndrome and IgAN, with one of the patients also having autoimmune hepatitis. The clinical state of the patient with autoimmune hepatitis was improved after the use of methylprednisolone, polyene phosphatidylcholine, and ursodeoxycholic acid [59].

Singhal and Sharma have also described a pediatric case of autoimmune hepatic disease in which mesangial IgA deposits were found. The patient experienced proteinuria and hematuria and eventually stage 3 chronic kidney disease [132].

### 8.6. Cirrhosis

Increased production or impaired clearance of Ig in individuals with liver cirrhosis may result in hypergammaglobulinemia, which may play a role in the development of secondary IgAN [9,133]. Hepatocytes’ asialoglycoprotein receptor binds desialylated glycoproteins via recognizing glycans with terminal galactose or N-acetylgalactosamine, and this appears to be the major IgA clearance pathway. The causes of aberrant glycosylated IgA1 overproduction are unknown; however, they may be related to pathogenic bacterium translocation, especially in cirrhotic individuals [134]. Cirrhosis impairs homeostasis on a number of linked levels, including pathological bacterial translocation, increased intestinal permeability, and aberrant antimicrobial peptide synthesis and clearance. Pathogenic bacteria translocation causes a proinflammatory milieu in the mucosal system in individuals with cirrhosis due to a disrupted intestinal barrier. This setting may cause a substantial amount of abnormally glycosylated IgA1 to be produced.

Yousaf et al. presented a case of a 72-year-old female with IgAN secondary to liver disease. The patient’s history included primary biliary cirrhosis and right knee arthroplasty with one month of post-operative prophylaxis against deep venous thrombosis (DVT) with warfarin. Due to the appearance of hematuria, a renal biopsy was carried out. The results indicated nephropathy induced by the warfarin as well as IgA deposits. The authors noted that the secondary IgAN was caused by warfarin nephrotoxicity [135].

In conclusion, it is well known that hepatic conditions can precede sIgAN in both pediatric and adult populations. While the pathomechanism linking liver cirrhosis and sIgAN has been studied, further research is required to fully identify the cause of glycosylated IgA1 overproduction in the case of liver cirrhosis. With regard to the other hepatic conditions, there is a lack of understanding of the definitive etiology of how sIgAN arises.

## 9. IgA Nephropathy Secondary to Iatrogenic Causes

IgAN may be induced by various medications [136]. This is suggested by the timely association between the administration of the pharmacological agent and new-onset nephropathy in cases with no history of previous kidney disease, by a relapse of IgAN following the repeated administration of the drug, improvement of pathologic changes upon drug withdrawal, and exclusion of other possible etiologies [136,137].

### 9.1. Tumor Necrosis Factor (TNF)-α Inhibitors

The majority of cases are connected to tumor necrosis factor (TNF)-α inhibitors, as they play a distinct role in modulating the immune system and because of their broad spectrum of use. Their side-effect profile includes the development of asymptomatic hematuria and proteinuria. The resultant pathologic characteristics are identical between primary and sIgAN.

A 61-year-old man with chronic plaque psoriasis was started on adalimumab, a fully humanized monoclonal TNF α inhibitor. He had no history of renal disease, and previous renal function tests were unremarkable. When renal failure became apparent on routine blood tests, his dermatologist referred him to a renal clinic, to which he presented 18 months after starting therapy with adalimumab. A renal biopsy revealed IgA mesangioproliferative glomerulonephritis, which prompted the discontinuation of adalimumab. After three weeks, his laboratory values began to improve continuously. That suggests a causal relationship between the drug and pathological changes. Furthermore, his IgAN is not likely a result of his psoriasis as it is well controlled [138].

A possible explanation for the pathological process involved in the development of IgAN secondary to the use of TNF-α inhibitors is, in addition to the genetic predisposition to the presence of aberrantly glycosylated IgA1, which is recognized by IgG autoantibodies, the partaking of antibodies against the glycan structure found in the heavy chains of TNF-α inhibitors. Another possible process suggests the involvement of aberrantly glycosylated IgA1 binding to the antigenic epitopes of TNF-α inhibitors. Finally, large polymeric IgA complexes are formed, which deposit in the mesangium [136]. The immune complexes then trigger the complement system and promote switching from T-helper type 1 to type 2 cytokine response, amplifying antibody production [138].

Infliximab, which belongs to the group of TNF-α inhibitors, was initiated in the case of a 25-year-old woman with generalized pustular psoriasis. She had episodic gross hematuria during this treatment, a gradual rise in proteinuria, and renal dysfunction related to upper respiratory tract infections. Her treatment with infliximab was terminated after the second cycle, which was three years after the initial administration of infliximab. One month later, a kidney biopsy revealed IgAN. She was started on corticosteroids, later substituted by secukinumab, an anti-IL-17A monoclonal antibody. In addition, a bilateral tonsillectomy was performed, and her laboratory tests improved. TNF-α inhibitors may induce IgAN. This clinical scenario exacerbated the patient’s IgAN, which developed secondary to her generalized pustular psoriasis. The connection between the pharmacological agent and the sIgAN is indicated by the deteriorated urinalysis with upper respiratory infection following the commencement of infliximab administration [46].

Another case describes a 56-year-old man with severe psoriasis and peripheral arthritis who received infliximab. After the fifth cycle, microscopic hematuria and proteinuria appeared, which prompted a kidney biopsy. He was diagnosed with IgAN, and therapy with ustekinumab, an interleukin (IL)-12/IL-23-inhibiting monoclonal antibody, was initiated, and his condition improved. Consequently, a renal biopsy was performed, and IgAN was recognized. He was started on infliximab, and after the second cycle, his conditions ameliorated together with a reduction in erythrocyte sedimentation rate and proteinuria. The patient has since not had any exacerbation of his renal disease [139].

The other case presents a 46-year-old man with psoriasis and oligoarthritis. Later, a renal biopsy revealed IgAN, which presented with hematuria. Initially, he received infliximab as monotherapy, which improved his conditions and led to a decrease in proteinuria. However, once the response to infliximab was reduced, methotrexate and cyclosporine were added to his treatment regime, which improved his state. In both events, the renal condition improved upon control of the comorbidities. This suggests that in some cases, TNF-α inhibitors may be beneficial. The authors discussed that such pharmacological agents could reduce cytokine production and inhibit macrophage migration inhibitory factor (MIF), which plays a crucial role in immunological kidney damage [46].

### 9.2. IL-12/IL-23-Inhibitor

On routine evaluation, a 24-year-old man with previously normal urinalysis was diagnosed with Crohn’s disease (CD). In the same year, an abnormal urinalysis was detected incidentally. His initial therapy was changed to ustekinumab. After the second administration of ustekinumab, the patient’s renal function deteriorated, while his gastrointestinal symptoms were absent. A renal biopsy identified IgAN related to aberrant IgA1 formation due to ustekinumab. Ustekinumab therapy was discontinued, and steroids were prescribed instead, which yielded improved renal function. IgAN did not improve, contrary to Crohn’s disease. Hence, the authors assume that the patient suffered from primary IgAN exacerbated by ustekinumab. In this case, blocking the IL-12/23 signaling by ustekinumab may have promoted the production of faulty IgA1 due to a relative type 2 T-helper cell dominance. This, in turn, favors the development of IgAN [26].

According to these opposing findings, further research is necessary in order to identify all factors relevant to the complex interaction of sIgAN and certain medications [44].

### 9.3. Immune Checkpoint Inhibitors

A study conducted at The University of Texas MD Anderson Cancer Center reported two patients with immune checkpoint inhibitor (CPI)-related IgAN. One patient developed IgAN following a combined treatment of ipilimumab and nivolumab; the other received pembrolizumab monotherapy. Neither one had a medical history of IgAN. The patient receiving the combination therapy had chronic kidney disease (CKD) stage 3 prior to receiving his treatment. His laboratory values deteriorated after the administration of the second round of medications, and a kidney biopsy revealed IgAN. The other patient’s kidney function deteriorated after the fifth cycle of pembrolizumab, and a diagnosis of IgAN was made based on a kidney biopsy. Both patients had partial improvement of their conditions following treatment with steroids and, in one case, additionally mycophenolate mofetil and infliximab. As CPIs induce TNF-alpha, infliximab might have played a crucial role in the treatment of the refractory case [140].

Nivolumab, a fully human IgG4 antibody belonging to the immune checkpoint inhibitor (ICI) group, acts by inhibiting anti-programmed cell death 1 (PD-1). PD-1 is a transmembrane protein found on the surface of T cells, B cells, and natural killer cells [11]. The ability of ICIs to potentiate immune responses may lead to immune-related adverse events involving the kidneys [141]. A 72-year-old man received nivolumab for the treatment of recurrent lung squamous cell carcinoma after an operation. He had no previous medical history of abnormal urinalysis or renal dysfunction. Evaluations of urinalysis obtained prior to the treatment with nivolumab were unremarkable. Half a year after the commencement of nivolumab therapy, the laboratory kidney values worsened, and the patient developed proteinuria.

A kidney biopsy was performed and revealed IgAN. Upon discontinuation of nivolumab, the laboratory values improved and were maintained. The condition of the patient improved once nivolumab therapy was ceased. Because of the lack of other potential triggers, the authors presumed that nivolumab’s inhibitory action regarding the PD-1 signaling pathway may have aggravated IgA formation [142].

Another case involving nivolumab concerns a 78-year-old man diagnosed with advanced gastric cancer with portal thrombosis and underlying type 2 diabetes mellitus, which was well controlled. Urinalysis prior to nivolumab therapy displayed trace proteinuria. While the treatment was effective in regard to his cancer, he quickly developed signs of impaired kidney function, and a kidney biopsy demonstrated IgAN connected to nivolumab. Nivolumab therapy was stopped, and the patient received steroids which improved his renal condition [141].

An additional case mentioned a 70-year-old man diagnosed with metastatic clear cell renal cell carcinoma. The presence of glomerular deposits in the kidney biopsy suggested that post-infectious glomerulonephritis might have occurred. However, prior to hospitalization, the patient had neither pharyngitis nor a skin infection. It was also incongruous with post-infectious glomerulonephritis to have neither hypertension nor a normal C3 level. Initially, his body reacted well to treatment with nivolumab (left and right kidneys reduced by 19 and 13%, respectively, and adrenal masses decreased by 23% on both sides). However, after eight months, he presented with acute kidney injury (AKI). A kidney biopsy was obtained, and nivolumab-induced immune complex-mediated glomerulonephritis was suspected, which urged the discontinuation of the current treatment. The patient was administered steroids and underwent hemodialysis leading to renal function stabilization [143].

### 9.4. CTLA4-Ig

A 47-year-old woman with rheumatology was started on abatacept monotherapy after previous treatments failed. Abatacept, a cytotoxic T-lymphocyte antigen 4 (CTLA4) Ig, is a recombinant human fusion protein that binds to CD80/86. This reaction modifies the binding of CD80/86 to CD28 and allows inhibition of T-cell activation. The patient routinely underwent urinalyses which demonstrated microscopic hematuria and worsening proteinuria. Abatacept was discontinued; however, proteinuria remained, and a kidney biopsy revealed IgAN. Steroid treatment was initiated, and the kidney damage improved. While the temporal connection between abatacept and the incidence of glomerulonephritis and its immediate improvement after ceasing the treatment suggests that the IgAN is secondary to the drug, there may have been a delayed adverse effect of adalimumab with which the patient was treated prior to changing to abatacept [144].

### 9.5. Oral Anticoagulants

Rivaroxaban is a factor Xa direct inhibitor used in the therapy of venous thromboembolism, as in the case of a 45-year-old man with pulmonary emboli and deep vein thrombosis. His past medical history involved asthma. A week after commencing therapy with rivaroxaban, he developed a non-blanching purpuric rash, night sweats, myalgias, arthralgias, pitting edema of the legs, nausea, and bilateral flank pain. Rivaroxaban was switched to apixaban, and the symptoms improved. Laboratory tests revealed proteinuria, microscopic hematuria, and red cell casts. A renal biopsy was performed because of persistent proteinuria and urinary sediment activity, and a diagnosis of IgAN was made. The authors suggest that rivaroxaban may be involved in immune complex formation and deposition, leading to IgAN [145].

In the case of a 61-year-old man, no proteinuria or microscopic hematuria was present prior to treatment with dabigatran, a direct oral anticoagulant. After one year of therapy, he presented with gross hematuria in the setting of acute kidney injury. His serum creatinine level increased significantly, and dabigatran was stopped. He was started on vitamin K and put on temporary hemodialysis. His symptoms improved, and after complete clearance of gross hematuria, a kidney biopsy revealed IgAN [146].

Along with rivaroxaban and dabigatran, anticoagulant-related nephropathy has also been associated with the use of vitamin K antagonists—acenokumarol and warfarin [147]. Ishii et al. reported a case of a 55-year-old man who developed acute kidney injury as a result of taking warfarin for anticoagulant therapy. In this case, the patient presented with gross macrohematuria, mild edema, and a serum creatinine level of 9.01 mg/dL. Additionally, urinalysis revealed proteinuria (0.95 g/gCr). The levels of proteinuria and hematuria were reduced after the infusion of extracellular fluid solution [148].

### 9.6. Anti-Vascular Endothelial Growth Factor

A 68-year-old man with metastatic rectal cancer was treated with bevacizumab, a monoclonal antibody for vascular endothelial growth factor. It was suggested that lower levels of VEGF directly affect the glomerular endothelium, and bevacizumab is known to cause proteinuria.

Throughout this therapy, he developed hematuria as well as proteinuria. A biopsy was performed and revealed IgAN. After cessation of bevacizumab, proteinuria almost completely resolved, and a follow-up renal biopsy showed a decrease in IgA deposits. A year later, the follow-up demonstrated cessation of hematuria [149].

### 9.7. Thioureylene Derivative

An 11-year-old girl with Graves’ disease was given propylthiouracil (PTU), which is a thioureylene derivative. She developed microscopic hematuria and proteinuria, and a renal biopsy was performed which demonstrated IgAN; however, a possible PTU-induced pauci-immune glomerulonephritis was also considered. PTU was discontinued and the patient received steroid therapy. The steroid dose was reduced following the improvement of her creatinine value, and cyclophosphamide therapy was initiated. Eventually, cyclophosphamide was discontinued, and a follow-up biopsy was consistent with only IgAN. Her treatment was adjusted by tapering prednisone and administering fish oil (Maxepa) and enalapril, as the patient was still presenting with proteinuria. Upon her last check-up, her urinalysis demonstrated lower levels of proteinuria. The authors are not certain whether this is a case of IgAN secondary to the use of PTU or if the patient had preexisting IgAN and developed pauci-immune GN secondary to the medication [150] (Table 2).

## 10. IgA Nephropathy Secondary to Environmental Exposure

Silica is one of the most abundant minerals on earth. It induces the activation of macrophages, monocytes, and lymphocytes and causes dysregulation of the immune system [66]. The following cases describe scenarios of IgAN secondary to occupational silica exposure.

A 43-year-old man with silicosis presented to the clinic with bilateral edema of the lower extremities and recent onset of gross hematuria. He was diagnosed with a lower limb varicose vein. He worked for 30 years as a coal miner. A kidney biopsy was obtained because of worsening signs of the impaired renal condition, which showed IgAN with acute tubulointerstitial nephritis. The authors proposed that the patient could have sIgAN associated with silicosis. This suggestion is supported by the dominance of NLRP3-mediated inflammation on biopsy, which was proposed to be associated with silicosis in this patient. The possible pathomechanisms involve either a direct noxious effect of the crystalline substance in the kidneys or an autoimmune reaction between silica and the immune system [65].

Another case of a patient with occupational exposure to silica describes a 26-year-old man who worked as a stonemason for five years, during which he mostly neglected to wear protective clothing. He was found to have hematuria and proteinuria. A renal biopsy revealed IgAN. He was started on steroids due to findings suggesting the presence of antineutrophil cytoplasmic antibody (ANCA)-associated vasculitis (AAV). In addition, a change in occupation was suggested. Three months after the initiation of his treatment, serum creatinine, as well as hematuria, improved. This case suggests a causal relationship between silica and the elicited immune dysregulation, resulting in IgAN and severe AAV. It is also possible that the patient already had IgAN, which would have been associated with silica-induced AAV in this clinical scenario.

In the case of a 51-year-old man with silicosis and hematuria, a biopsy confirmed the diagnosis of IgAN. After he stopped working as a building wrecker, his condition improved, including urinalysis. Given the above, a thesis on the link between silica exposure and IgAN was supported [66].

The suggested pathomechanism of silica exposure leading to sIgAN revolves around the silica-induced overactivation of the immune system or its direct structural impact on the kidneys [65]. It is advised to eliminate the causative agent. Further studies are needed to determine the exact pathomechanism to achieve more efficient treatment [66].

## 11. Conclusions

We identified a multitude of medical conditions that can predispose to sIgAN development. Nephritic syndrome and hematuria were the most common manifestations of sIgAN. The basis of the treatment strategy was proper therapy of the underlying condition.

Regarding drug-induced IgAN, discontinuation alone was insufficient to improve renal symptoms. In non-oncological cases, antimicrobial or anti-inflammatory approaches following renin–angiotensin-system blockade were utilized. Occupational silica exposure causing sIgAN was also described. The majority of patients reported improvement or reached partial remission, with only a few cases achieving full recovery, especially when the exogenous factor was identified and avoided.

Due to the shared pathogenesis of pIgAN and sIgAN involving glomerular IgA1 with galactose deficiency of O-glycans, it seems reasonable to conduct studies concerning genetic predisposition, as was undertaken for primary IgAN, and to elucidate the role of microbiota in sIgAN.

Considering all of the above, clinicians should always be aware of the possibility of the occurrence of sIgAN in patients with listed primary diseases. That knowledge may also be essential while assessing patients with IgAN for secondary causes. Such management is recommended following Kidney Disease: Improving Global Outcomes (KDIGO) guidelines [151].

## 12. Limitations

Despite the abundance of presented putative sIgAN causes, we do believe that dozens of other factors remain to be determined, especially concerning infectious diseases. The explanation for such underestimation is limited literature on sIgAN caused by the low prioritization of case reports by researchers, publishing policy focused on original papers, economic issues, and underdiagnosing IgAN. The sIgAN diagnosis is established by examining renal tissue after a biopsy and immunofluorescence or immunohistochemical staining. That procedure is invasive and not performed routinely. Ordinarily, empirical non-specific anti-inflammatory treatment is commissioned with positive results. Another reason for sIgAN causes being under-reported is a self-limiting disease course in many cases. Furthermore, case report studies usually do not comprise long-term follow-up. Thus, response to treatment was not always closely depicted. Some articles lacked details on the IgAN phenotype. 

## Figures and Tables

**Figure 1 jcm-12-02726-f001:**
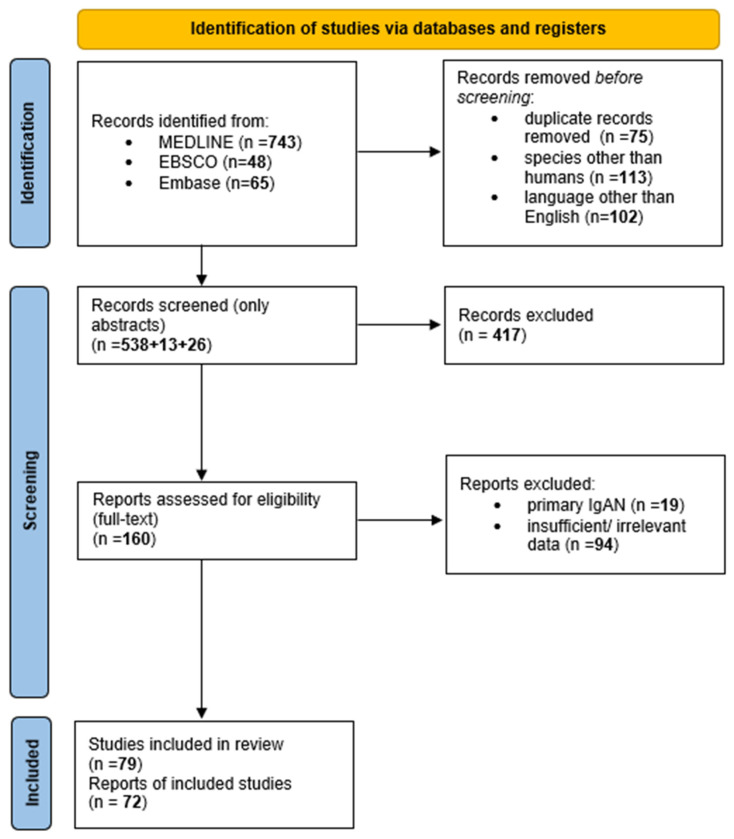
A flow diagram based on PRISMA guidelines [17].

**Table 1 jcm-12-02726-t001:** Summary of case reports on sIgAN and IgA-associated nephropathy.

Primary Disease	IgAN Phenotype	No. of Case Reports	Introduced Treatment	Response to Treatment	References
Crohn’s disease	subnephrotic proteinuria, microscopic hematuria	6	mesalazine, prednisolone, AZA, infliximab; ileum resection	remission	Terasaka et al. [23]
	subnephrotic proteinuria, microscopic hematuria		mesalazine, loperamide, deflazacort, cyclophosphamide → AZA → mycophenolate sodium; right hemicolectomy	AZA caused bone marrow suppression; after four months of treatment with mycophenolate sodium and deflazacort, renal functions were maintained	Choi et al. [24]
	nephrotic proteinuria, gross hematuria		mesalazine, methylprednisolone, adalimumab (potentially triggered IgAN), vedolizumab	remission	Mertelj et al. [25]
	subnephrotic proteinuria, microscopic hematuria		infliximab → ustekinumab (UST) due to therapy-resistant diarrhea and bloody stools (UST potentially triggered IgAN), prednisolone, methylprednisolone, losartan	improvement	Kanazawa et al. [26]
	gross hematuria		oral steroids, right hemicolectomy, and resection of the terminal ileum	IgAN improvement only after surgery	Forshaw et al. [27]
	subnephrotic proteinuria, gross hematuria		methylprednisolone, AZA	improvement	Pipili et al. [28]
Ulcerative colitis	subnephrotic proteinuria, microscopic hematuria	4	metoprolol, methimazole (potentially triggered IgAN and ANCA-vasculitis), prednisone, mesalazine; intravenous steroids	improvement	Ku et al. [29]
	subnephrotic proteinuria, microscopic hematuria		sulfasalazine: IgA-associated nephropathy required no medication	improvement	Iida et al. [30]
	subnephrotic proteinuria, gross hematuria		resection of ileorectal pouch; IgA-associated nephropathy required no medication	improvement of UC and IgAN; hematuria and proteinuria gradually decreased without IgAN therapy	Onime et al. [31]
	gross hematuria		mesalazine	improvement, microscopic hematuria persisted	Trimarchi et al. [22]
Celiac disease	nephrotic proteinuria, microscopic hematuria	5	large doses of furosemide and metolazone; gluten-free diet	improvement	Woodrow et al. [32]
	nephrotic proteinuria		ACE inhibitor, oral iron; gluten-free diet	remission	Habura et al. [33]
	nephrotic proteinuria, microscopic hematuria		enalapril, valsartan, prednisone, azathioprine → mycophenolate; gluten-free diet	improvement	Koivuviita et al. [34]
	subnephrotic proteinuria		ramipril, prednisolone, mycophenolate sodium, dapsone; gluten-free diet	initial improvement after a gluten-free diet; six months later: proteinuria relapse, dermatitis herpetiformis development, septicemia leading to death	Gupta et al. [35]
	subnephrotic proteinuria, microscopic hematuria		gluten-free diet	improvement	La Villa et al. [36]
*C. difficile* colitis	subnephrotic proteinuria, gross hematuria	1	vancomycin; methylprednisolone, cyclophosphamide	remission	Gaughan et al. [37]
HIV infection	subnephrotic proteinuria, microscopic hematuria	5	raltegravir, abacavir, lamivudine, pulse following oral steroid therapy; tonsillectomy	improvement	Kawakita et al. [38]
	nephrotic proteinuria, gross hematuria		the patient refused antiretroviral therapy, prednisone	improvement	Hsieh et al. [39]
	nephrotic proteinuria, microscopic hematuria		didanosine, captopril	improvement	Gorriz et al. [40]
	subnephrotic proteinuria, microscopic hematuria		methylprednisolone	improvement: proteinuria and microscopic hematuria persisted	Jindal et al. [41]
	nephrotic proteinuria, microscopic hematuria		antiretroviral therapy, 3x methylprednisolone pulses, oral prednisolone therapy; tonsillectomy	improvement	Miyasato et al. [42]
Psoriasis	proteinuria, hematuria	6	secukinumab, tonsillectomy	remission	Ochi et al. [43]
	proteinuria, microscopic hematuria,		infliximab	remission	Sakellariou et al. [44]
	proteinuria, hematuria		infliximab, methotrexate, cyclosporin	improvement	
	nephrotic syndrome, microscopic hematuria		diuretics, antihypertensive, hemodialysis, cyclophosphamide, steroids	improvement	Manchanda et al. [45]
	proteinuria, gross hematuria		steroids, tonsillectomy, IL17-α inhibitor	improvement	Segawa et al. [46]
	nephrotic proteinuria		prednisone	improvement	Zadrazil et al. [47]
Dystrophic epidermolysis bullosa—recessive	rapidly progressive GN	1	steroids in the pre-dialysis period, then dialysis and kidney transplantation	deteriorated to ESRD (steroid-resistant GN)	Ceuppens et al. [48]
Epidermolysis bullosa—dominant	proteinuria, hematuria	1	steroids, mizoribine, dilazep dihydrochloride, warfarin	improvement	Kawasaki et al. [49]
Non-Herlitz junctional epidermolysis bullosa	rapidly progressive GN, nephrotic proteinuria	1	unknown	deteriorated to ESRD, kidney transplant performed	Ungureanu et al. [50]
Alcoholic cirrhosis	(a), (b) proteinuria and hematuria	4	steroids	(a), (b) proteinuria resolved	Takada et al. [51]
	(a) rapidly progressive nephritic syndrome, gross hematuria (b) subnephrotic proteinuria		(a) bilateral tonsillectomy, steroids, mizoribine (b) steroids and mizoribine	(a) resolution of hematuria, proteinuria, improvement of renal function (b) proteinuria and serum creatinine decreased	Kaneko et al. [52]
Wilson’s Disease	microscopic hematuria, proteinuria, renal dysfunction	2	trientere HCl, zinc acetate	renal manifestations improved	Shimamura et al. [53]
	proteinuria, gross hematuria		D-penicillamine	hematuria resolved; proteinuria reduced	Bhandari et al. [54]
Hemochromatosis	proteinuria, gross hematuria	1	steroids	improvement in renal function; however, due to a cerebral hemorrhage, steroids were discontinued; cardiopulmonary arrest 10 days later	Nakayama et al. [55]
Hepatitis	GFR decline	5	conservation management, sofosbuvir/velpatasvir	improvement	Denha et al. [56]
	proteinuria		ribavirin, pegylated INF-α	reduction of proteinuria	Dey et al. [57]
	proteinuria, hematuria		ribavirin and INF- α	resolution	Ji et al. [58]
	(a) proteinuria, macroscopic hematuria (b) microscopic hematuria		(a) methylprednisolone, polyene phosphatidylcholine, ursodeoxycholic acid, methotrexate (b) methylprednisolone, methotrexate	(a) progression stopped (b) persistence of microscopic hematuria	Chen et al. [59]
Sjorgen syndrome	nephrotic syndrome, microscopic hematuria, vasculitis	1	glucocorticoids and treatment of local symptoms (ocular and oral dryness); hemodialysis	despite receiving hemodialysis, the patient’s renal function showed no improvement	Tsai et al. [60]
Spondyloarthritis	subnephrotic proteinuria, microscopic hematuria	1	irbesartan	renal function and proteinuria remained stable	Jacquet et al. [61]
Renal cell carcinoma	subnephrotic proteinuria, microscopic hematuria	3	nephrectomy	after nephrectomy, proteinuria and hematuria were decreased or resolved	Mimura et al. [62]
Gastric adenocarcinoma	nephrotic syndrome, microscopic hematuria	1	methylprednisolone, losartan	improvement	Kocyigit et al. [63]
Hodgkin’s lymphoma	hematuria, elevated creatinine	1	chemotherapy	after 8th cycle of chemotherapy, renal function improved; however, patient died due to ARDS	Yoo et al. [64]
Silicosis	nephrotic proteinuria, gross hematuria	1	steroids, ACE inhibitors	improvement	Chen et al. [65]
Silica exposure ANCA-associated vasculitis	subnephrotic proteinuria, gross hematuria	1	steroids, cyclophosphamide	improvement	Rao et al. [66]
Henoch–Schönlein purpura	nephrotic proteinuria, microscopic hematuria	1	PA	after 4th PA, creatinine levels were stable	Lee et al. [67]
MRSA infection	nephrotic proteinuria, gross hematuria	1	vancomycin, rifampin, ciprofloxacin, linezolid	improvement	Riley et al. [68]
MSSA infection	nephrotic proteinuria, no hematuria	1	methylprednisolone	improvement	Javed et al. [69]
*M. pneumoniae* infection	subnephrotic proteinuria, gross hematuria	2	erythromycin	remission	Kanayama et al. [70]
	subnephrotic proteinuria, gross hematuria		minocyclin, steroids, furosemide, catecholamine, clarithromycin	improvement	Suzuki et al. [71]
Lyme disease	new proteinuria, gross hematuria	1	steroids, doxycycline	improvement	McCausland et al. [72]
Cat scratch disease	subnephrotic proteinuria, gross hematuria	1	amoxicillin/clavulanic acid, azithromycin, rifampin	improvement	Hopp et al. [73]
Osteomyelitis	subnephrotic proteinuria, microscopic hematuria	1	amputation	improvement	Tevlin et al. [74]
Tonsillitis	hematuria	1	tonsillectomy	remission	Liess et al. [75]
Malaria	subnephrotic proteinuria, microscopic hematuria	1	quinine dihydrochloride, doxycycline, hemodialysis	improvement	Yoo et al. [76]
Schistosomiasis	nephrotic proteinuria, no hematuria	1	losartan, atenolol	improvement	Gonçalves et al. [77]

Mesalazine—5-aminosalicylic acid/5-ASA/mesalamine; AZA—azathioprine; UC—ulcerative colitis; UST—ustekinumab; PA—plasmapheresis; subnephrotic proteinuria—proteinuria <3.5 g/day; nephrotic proteinuria—≥3.5 g/day; gross hematuria—macroscopic hematuria/visible blood in the urine; microscopic hematuria—red blood count/no visible blood in the urine; GN—glomerulonephritis; ESRD—end-stage renal disease; ACE—angiotensin-converting enzyme; HCl—hydrochloric acid.

**Table 2 jcm-12-02726-t002:** Summary of literature concerning drug-induced IgA nephropathy.

Medication	Primary Disease/Diseases	IgAN Phenotype	Suggested Treatment	Remission after Withdrawal	Reference
**TNF-α inhibitors**					
**Infliximab**	ankylosing spondylitis	nephrotic proteinuria, microscopic hematuria	steroids, ACE inhibitors	partial remission	Diena et al. [137]
**Infliximab**	pustular psoriasis	subnephrotic proteinuria, gross hematuria	steroids, secukinumab, bilateral tonsillectomy	partial remission	Segawa et al. [46]
**Infliximab**	psoriasis, peripheral arthritis	subnephrotic proteinuria, microscopic hematuria	ACE inhibitors, ustekinumab	partial remission	Kluger et al. [139]
**Adalimumab**	plaque psoriasis	subnephrotic proteinuria, microscopic hematuria	steroids, amlodipine candesartan + hydrochlorothiazide lincosa	remission	Wei et al. [138]
**Immune checkpoint inhibitors**					
**Pembrolizumab**	HT, GERD, asthma	no proteinuria, 2 RBC/HPF	steroids, mycophenolate mofetil, infliximab	partial remission	Mamlouk et al. [140]
**Nivolumab + Ipilimumab**	HT, GERD, CKD stage 3.	nephrotic proteinuria 11 RBC/HPF	steroids	remission	Mamlouk et al. [140]
**Nivolumab**	lung squamous cell carcinoma	subnephrotic proteinuria, gross hematuria	N.D.	partial remission	Kishi et al. [142]
**Nivolumab**	gastric cancer, portal thrombosis, DM type 2.	nephrotic proteinuria, gross hematuria	steroids	partial remission	Tanabe et al. [141]
**Nivolumab**	metastatic clear cell renal cell carcinoma	subnephrotic proteinuria, gross hematuria	steroids, hemodialysis	partial remission	Jung et al. [143]
**CTLA4-Ig**					
**Abatacept**	rheumatoid arthritis	subnephrotic proteinuria, microscopic hematuria	steroids	partial remission	Michel et al. [144]
**Oral anticoagulants**					
**Dabigatran**	HT, alcoholic liver cirrhosis paroxysmal atrial fibrillation	subnephrotic proteinuria, gross hematuria	vitamin K, hemodialysis, steroids	remission	Li et al. [146]
**Rivaroxaban**	pulmonary emboli, DVT	nephrotic proteinuria, microscopic hematuria	apixaban ramipril	partial remission	Chung et al. [145]
**Warfarin**	Marfan syndrome, aortic valve regurgitation	subnephrotic proteinuria	infusion of extracellular fluid solution	partial remission	Ishii et al. [148]
**IL-12/IL-23-inhibitor**					
**Ustekinumab**	Crohn’s disease	subnephrotic proteinuria, microscopic hematuria	steroids losartan	partial remission	Kanazawa et al. [26]
**Anti-vascular endothelial growth factor**					
**Bevacizumab**	metastatic rectal cancer	nephrotic proteinuria, microscopic hematuria	cessation of bevacizumab	improvement	Yahata et al. [149]
**Thioureylene derivative**					
**Propylthiouracil**	Graves’ disease	nephrotic proteinuria, microscopic hematuria	prednisolone, cyclophosphamide, fish oil (Maxepa), enalapril	improvement	Winters et al. [150]

Subnephrotic proteinuria—proteinuria <3.5 g/day; nephrotic proteinuria—≥3.5 g/day; gross hematuria—macroscopic hematuria/visible blood in the urine; microscopic hematuria—red blood count/no visible blood in the urine; HT—hypertension; ACE—angiotensin-converting enzyme; GERD—gastroesophageal reflux disease; CKD—chronic kidney disease; N.D.—no data; DM—diabetes mellitus; CTLA4-Ig—cytotoxic T lymphocyte-associated antigen-4-Ig; DVT—deep vein thrombosis; RBC/HPF—red blood cells per high power field.

## Data Availability

Data sharing is not applicable as no datasets were generated or analysed during the current study.

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
