# Peer review of "Secondary IgA Nephropathy and IgA-Associated Nephropathy: A Systematic Review of Case Reports"

_jcm, 2023, doi:10.3390/jcm12072726_

Round 1
Reviewer 1 Report
In this manuscript the authors review different causes of secondary IgA Nephropathy (sIgAN).
The review is interesting, and is well written. However, some important changes should be performance.
My main issue with the review is that, in contrast what the authors made in the section “gastrointestinal disease” and in “viral infections”, they describe thoroughly every reported case of SIgAN in the others aetiologies. The authors should resume the cases trying to get conclusions about them. Tables 2 and 3 give enough information about the individual cases.
In table 2 the authors use abbreviations “Azatioprina (AZA)” and plasmapheresis (P.A), that are later described in the section abbreviations. It is not necessary to repeat them. In the table 2, in the section ulcerative colitis, abbreviation U.C. is not described in the sections abbreviations. In the table 2, in the section Wilson’s disease, abbreviation HCl is not described in the sections abbreviations In the table 2 in the section Celiac disease the authors use the term doeses instead of doses. In the table 2 in the section psoriasis the letters a) b) must be eliminated.
There are errors in different references along the manuscript.
In the section bacterial infections, the authors should make references and talk about IgA-dominant postinfectious glomerulonephritis.
In the sections new oral anticoagulants, the authors really should talk about oral anticoagulation because there are described cases with warfarin (Kidney Int Rep. 2022 Jan 19;7(4):831-840).
Finally, a paragraph about differences in clinical presentation forms, histology and evolution of primary and secondary IgAN would be really interesting.
Author Response
Dear Editor of JCM,
Dear Reviewers,
We are sincerely grateful for revising our manuscript and recommendations that resulted in the improvement of the quality of our research paper.
Hereby, we present a detailed point-by-point response letter to every review. In comparison with the submitted manuscript, we excluded 1 reference (3. Vuong et al.) and added 7 new references.
We hope to have addressed Your concerns with these modifications:
REVIEWER #1
1."The authors should resume the cases trying to get conclusions about them"
Response: We focused on reducing the main text and making conclusions.
2."In table 2 the authors use abbreviations "Azatioprina (AZA)" and plasmapheresis (P.A), that are later described in the section abbreviations. It is not necessary to repeat them. In the table 2, in the section ulcerative colitis, abbreviation U.C. is not described in the sections abbreviations. In the table 2, in the section Wilson's disease, abbreviation HCl is not described in the sections abbreviations In the table 2 in the section Celiac disease the authors use the term doeses instead of doses. In the table 2 in the section psoriasis the letters a) b) must be eliminated. There are errors in different references along the manuscript."
Response: We are grateful for perceptiveness and corrected listed mistakes.
3."In the section bacterial infections, the authors should make references and talk about IgA-dominant postinfectious glomerulonephritis."
Response: It is obviously our mistake, we are grateful for pointing out. We added the "IgA-dominant postinfectious glomerulonephritis" section (lines 324-340).
4."In the sections new oral anticoagulants, the authors really should talk about oral anticoagulation because there are described cases with warfarin (Kidney Int Rep. 2022 Jan 19;7(4):831-840).
Response: We included a suggested review and added a case report on warfarin.
5."...a paragraph about differences in clinical presentation forms, histology and evolution of primary and secondary IgAN would be really interesting."
Response: We added more differences between pIgAN and sIgAN to the introduction. (lines 59-70).
Reviewer 2 Report
In this manuscript, the authors provide a review of the literature on diseases and drugs, that have been associated with IgA nephropathy.
I found the review well-written and interesting to read (albeit very long and I wonder if some sections could be less descriptive, in order to make it easier for the readers).
In the introduction (lines 76-77), you mention that you realized that the paper does not meet the systematic review definition, however, you kept the definition in the title. You should mention which criteria are lacking (eg PICOS framework, risk of bias assessment), be more specific, and elaborate on that sentence.
In the conclusion (line 874), you should rephrase the word 'mandatory' to 'is recommended'.
Author Response
Dear Reviewer,
We are sincerely grateful for revising our manuscript and recommendations that resulted in the improvement of the quality of our research paper.
Herein, we present a detailed point-by-point response letter to every review. In comparison with the submitted manuscript, we excluded 1 reference (3. Vuong et al.) and added 7 new references.
We hope to have addressed Your concerns with these modifications:
1."In the introduction (lines 76-77), you mention that you realized that the paper does not meet the systematic review definition, however, you kept the definition in the title. You should mention which criteria are lacking (eg PICOS framework, risk of bias assessment), be more specific, and elaborate on that sentence."
Response: (lines 100-103) our study lacked risk of bias assessment. Instead, we carried out a critical appraisal of the articles' quality by criteria similar to Joanna Briggs Institute Critical Appraisal Checklist for Case Reports. Two investigators checked the case reports' quality and those who did not meet the criteria were excluded and marked as "insufficient data".
2."In the conclusion (line 874), you should rephrase the word 'mandatory' to 'is recommended'."
Response: We are grateful for perceptiveness and corrected that sentence.
Reviewer 3 Report
The systematic review article by Tota M et al. on secondary IgA nephropathy contributes to the information on the clinical entities associated with glomerular IgA deposition, opening perspectives for understanding the pathophysiological mechanisms underlying the development of this glomerulonephritis.
Throughout the text, the term IgA nephropathy is used repeatedly in place of the acronym IgAN and this should be corrected.
The term nephritic does not apply to the level of proteinuria but rather to the presence of glomerular hematuria in the urinary sediment. Therefore the designation nephritic proteinuria, when it appears in the text, should be replaced by nephrotic proteinuria (lines 522 and 555).
The description of the clinical case referred to in line 765 should be revised, as it is not enlightening.
Author Response
Dear Reviewer,
We are sincerely grateful for revising our manuscript and recommendations that resulted in the improvement of the quality of our research paper.
Hereby, we present a detailed point-by-point response letter to every review. In comparison with the submitted manuscript, we excluded 1 reference (3. Vuong et al.) and added 7 new references.
We hope to have addressed Your concerns with these modifications:
"Throughout the text, the term IgA nephropathy is used repeatedly in place of the acronym IgAN and this should be corrected."
Response: We rephrased full terms with IgAN and sIgAN.
"The term nephritic does not apply to the level of proteinuria but rather to the presence of glomerular hematuria in the urinary sediment. Therefore the designation nephritic proteinuria, when it appears in the text, should be replaced by nephrotic proteinuria (lines 522 and 555)."
Response: We are grateful for perceptiveness and rephrased "nephritic" with "subnephrotic" proteinuria.
"The description of the clinical case referred to in line 765 should be revised, as it is not enlightening."
Response: We edited the following case (lines 782-793).
Round 2
Reviewer 1 Report
The manuscript has been improved and could be published